



# Validation of Aura MLS retrievals of temperature, water vapour and ozone in the upper troposphere and lower–middle stratosphere over the Tibetan Plateau during boreal summer

X. L. Yan[1,2], J. S. Wright[3], X. D. Zheng[1], N. Livesey[4], H. Vömel[5], and X. J. Zhou[1]

[1]Chinese Academy of Meteorological Sciences, Beijing, China
[2]Institute of Atmospheric Physics, Beijing, China
[3]Center for Earth System Science, Tsinghua University, Beijing, China
[4]Jet Propulsion Laboratory, California Institute of Technology, Pasadena, California, USA
[5]Earth Observing Laboratory, National Center for Atmospheric Research, Boulder, Colorado, USA

*Correspondence to:*
J. S. Wright (jswright@tsinghua.edu.cn)
X. D. Zheng (zhengxd@cams.cma.gov.cn)

**Abstract.** We validate Aura Microwave Limb Sounder (MLS) version 3 (v3) and version 4 (v4) retrievals of summertime temperature, water vapour and ozone in the upper troposphere and lower–middle stratosphere (UTLS; 10–316 hPa) against balloon soundings collected during the Study of Ozone, Aerosols and Radiation over the Tibetan Plateau (SOAR-TP). Mean v3 and v4 profiles of temperature, water vapour and ozone in this region during the measurement campaigns are almost

5 identical through most of the stratosphere (10–68 hPa), but differ in several respects in the upper troposphere and tropopause layer. Differences in v4 relative to v3 include slightly colder mean temperatures from 100–316 hPa, smaller mean water vapour mixing ratios in the upper troposphere (215–316 hPa), and a more vertically homogeneous profile of mean ozone mixing ratios below the climatological tropopause (100–316 hPa). These changes substantially improve agreement between ozonesondes and MLS ozone retrievals in the upper troposphere, but slightly worsen existing cold and dry biases in the upper troposphere.

Aura MLS v3 and v4 temperature profiles contain significant cold biases relative to collocated temperature measurements in several layers of the lower–middle stratosphere (mean biases of −1.3 to −1.8 K centered at 10–12 hPa, 26–32 hPa and 68–83 hPa) and in the upper troposphere (mean biases of approximately −2.3±0.3 K in v3 and −2.6±0.4 K in v4 between 147 and 261 hPa). MLS v3 and v4 profiles of water vapour volume mixing ratio generally compare well with collocated measurements, with a slight dry bias (v4: −8±4%) near 22–26 hPa, a slight wet bias (v4: +12±5%) near 68–83 hPa, and a more substantial

dry bias (v4: −32±11%) in the upper troposphere (121–261 hPa). MLS v3 and v4 retrievals of ozone volume mixing ratio are biased high relative to collocated ozonesondes through most of the stratosphere (18–83 hPa), but are biased low at 100 hPa. The largest positive biases in ozone retrievals are located at 83 hPa (approximately +70%); this peak was not identified by earlier validations and may be regionally or seasonally specific. Ozone retrievals are substantially improved in v4 relative to v3, with smaller biases in the tropopause layer, reduced variance below 68 hPa, larger data yields, and smoother gradients in the vertical

profile of ozone biases in the upper troposphere.





# 1 Introduction

Variations in temperature, water vapour and ozone in the upper troposphere and lower–middle stratosphere (UTLS) play critical roles in the Earth's radiation budget (Manabe and Wetherald, 1967), with important implications for climate change (Soden et al., 2008; Solomon et al., 2010; Dessler et al., 2013). Accurate observations of these variables at UTLS altitudes are difficult to obtain. Instruments mounted on balloon sondes and aircraft can be used to collect measurements with high resolution along specific flight tracks, but suffer from limited spatial and temporal coverage. Reanalyses and other data assimilation systems provide global coverage at frequent intervals, but are heavily influenced by the underlying numerical model and often do not assimilate observations of ozone or stratospheric water vapour. Satellite observations occupy something of a middle ground between these, with improved spatial and temporal coverage relative to sonde and aircraft measurements (at the expense of spatiotemporal resolution) and improved fidelity to the state of the atmosphere relative to reanalyses (at the expense of simultaneous global coverage).

Satellite retrievals are based on the mathematical processing and physical interpretation of observed atmospheric radiances, with errors and uncertainties that reflect imperfections in the design of the instrument used to conduct the observations and the algorithm used to process the observations. These errors can be globally systematic or vary with season and location, so that the evaluation and validation of satellite retrievals requires a geographically and temporally diverse set of independent validation measurements from a variety of observational and semi-observational platforms.

The Microwave Limb Sounder (MLS) instrument onboard the Aura satellite has provided near-continuous sun-synchronous observations of temperature, water vapour and ozone in the upper troposphere, stratosphere and lower mesosphere since August 2004 (Waters et al., 2006). Four versions of MLS data have been prepared for public release to date. The initial production version, version 1.5 (v1.5), was replaced by version 2.2/2.3 (v2) in 2007 and version 3.3/3.4 (v3) in 2010. The most recent production version, version 4.2 (v4), replaced v3 in February 2015.

The boreal summertime UTLS over the Tibetan Plateau is dominated by the Asian monsoon anticyclone. The composition and thermodynamic structure of this region differ substantially from other regions at this latitude and represent a mix of tropical and mid-latitude characteristics (Park et al., 2007). The Tibetan Plateau region has been variously described as "the world's water tower" (Xu et al., 2008) and an "ozone valley" (Zhou et al., 1995; Tobo et al., 2008), and the Asian monsoon anticyclone has been identified as a key pathway for the transport of water vapour and pollutants across the tropopause and into the global stratosphere (Fu et al., 2006; Randel et al., 2010; Wright et al., 2011; Ploeger et al., 2013). Biases in Aura MLS are well-characterized globally, with multiple validation analyses of temperature (Froidevaux et al., 2006; Schwartz et al., 2008), water vapour (Read et al., 2007; Lambert et al., 2007; Vömel et al., 2007a; Berthet et al., 2013; Hegglin et al., 2013; Hurst et al., 2014) and ozone (Jiang et al., 2007; Livesey et al., 2008; Tegtmeier et al., 2013) against radiosonde and ozonesonde networks, frostpoint hygrometers, ground-based lidars, aircraft data and other satellite retrievals. However, despite extensive use and analysis of MLS retrievals in the vicinity of the Asian monsoon anticyclone (e.g., Park et al., 2007; Uma et al., 2014), few previous studies have conducted focused validations of MLS observations in this region (with the notable exception of Yan et al., 2015, which is discussed further below). Moreover, few of the field observations used to evaluate MLS retrievals in





global validation studies have been collected in the vicinity of the Asian monsoon anticyclone. Characterization of biases in this region therefore relies on global and zonal mean comparisons with other satellite data sets.

Here, we present a validation of the third (v3) and fourth (v4) public releases of Aura MLS retrievals over the southeastern Tibetan Plateau and adjacent regions during boreal summer using balloon soundings collected at four sites over four different
5 years (Fig. 1 and Table 1) during the Study of Ozone, Aerosols and Radiation over the Tibetan Plateau (SOAR-TP). The results provide a detailed evaluation of MLS retrievals of temperature, water vapour and ozone in the vicinity of the Asian monsoon anticyclone. We describe the in situ data, key differences between v3 and v4, and the validation methodology in Section 2. We then present the results of the validation in Section 3 and discuss these results in the context of previous validation studies, related variables and differences between v3 and v4 in Section 4. We conclude with a summary of key findings.

## 2 Data and methodology

### 2.1 Sonde measurements

The balloon-borne sonde measurements used in this analysis were collected at four high-altitude locations in southwestern China (Fig. 1): Tengchong, Yunnan (August 2010); Naqu, Tibet (August 2011); Lhasa, Tibet (May–July 2012) and Linzhi, Tibet (June–July 2014). Table 1 lists geolocation information for each of these sites and the numbers of temperature, water vapour and ozone profiles from each site used in the validation analysis. Temperature measurements were collected using Vaisala RS80
(Tengchong and Naqu) and RS92 (Naqu, Lhasa and Linzhi) radiosondes. Profiles of water vapour mixing ratio were collected using cryogenic frost point hygrometer (CFH) instruments attached to RS80 (Tengchong and Naqu) and InterMet (Lhasa and Linzhi) radiosondes, as the RS92 radiosonde does not permit the attachment of a CFH. The InterMet radiosondes released at Lhasa and Linzhi were launched together with RS92 radiosondes, and are therefore directly comparable. Ozone measurements
were collected using electrochemical concentration cell (ECC) instruments. Profiles of temperature, water vapour and ozone were obtained up to the burst point of each balloon, which typically occurred at altitudes greater than 30 km and pressures as low as 5 to 10 hPa. Launch times varied, but were predominantly during the early afternoon local time at Tengchong, Naqu and Lhasa, and were predominantly around midnight local time at Linzhi.

The CFH is a microprocessor and chilled mirror instrument capable of measuring a large range of water vapour concentra-
25 tions from the surface to approximately 28 km altitude (Vömel et al., 2007b). Cryogenic fluid is used to maintain the mirror at the frost point temperature, which is then converted to water vapour mixing ratio using the approximation to the Clausius–Clapeyron relation proposed by Goff and Gratch (1946). The uncertainty in CFH measurements is less than 10% in the upper troposphere and stratosphere.

ECC ozonesondes observe ozone mixing ratios by measuring electrical currents produced by reactions of $O_3$ and potassium
iodide (KI) in separate cathode and anode chambers. These electrical currents are directly proportional to the amount of ozone in air pumped into the instrument. The minimum detection limit is approximately 2 ppbv, considerably less than the typical background value for clean tropospheric air (30 ppbv). ECC measurements are typically accurate to within 10% in the troposphere and 5% in the stratosphere up to 10 hPa (Smit et al., 2007).





During flight, the CFH and ECC data streams were transmitted to receiving equipment on the ground through interfaces for the RS80 (Tengchong and Naqu), InterMet (Lhasa and Linzhi) and RS92 (Naqu, Lhasa and Linzhi; ECC only) radiosondes. These data were stored together with profiles of pressure, temperature and other variables observed by the radiosonde instrument. The payloads weighed approximately 1 kg and were flown using 1600 g latex balloons filled with hydrogen. Although

only ascending data are analyzed here, each balloon was equipped with a parachute to enable the potential use of data collected during descent and recovery of the instrument package.

## 2.2 Aura MLS temperature, water vapour and ozone retrievals

Versions 3 and 4 of the MLS retrieval algorithm have been used to process the third and fourth public releases of MLS data, respectively (henceforth referred to as v3 and v4). Both versions of the data consist of profiles reported on 12 pressure levels per

10 decade between 1000 hPa and 1 hPa, 6 pressure levels per decade between 1 hPa and 0.1 hPa, and 3 pressure levels per decade between 0.1 hPa and 0.01 hPa. The MLS measurement system uses optimal estimation theory (Rodgers, 2000) to retrieve an atmospheric state vector (Livesey et al., 2013, 2015). Temperature profiles are retrieved using radiances near the $O_2$ spectral bands at 118 GHz (for the stratosphere and above) and 239 GHz (for the troposphere), water vapour profiles are retrieved using radiances at 190 GHz and ozone profiles are retrieved using radiances at 240 GHz. The atmospheric state vector produced by

15 the full retrieval algorithm contains estimates of temperature, water vapour and ozone at 55 pressure levels (as well as other variables that are not considered here). The profiles used in this validation analysis have been screened using the quality control criteria suggested by Livesey et al. (2013) for v3 and Livesey et al. (2015) for v4 (reproduced in Appendix A). We validate MLS profiles of water vapour, temperature and ozone at the 19 standard MLS pressure levels between 316 hPa and 10 hPa (see Fig. 2). Although MLS retrievals of temperature and ozone at pressures greater than 261 hPa are currently not recommended

for use in scientific studies, we evaluate and briefly discuss the performance of retrievals of both variables at 316 hPa.

Uncertainties in MLS measurements are estimated by combining the precisions of the radiance observations with uncertainties in the a priori estimates as described by Rodgers (1976). These uncertainty estimates represent the diagonal elements of the solution covariance matrix, and are provided for each profile in the MLS Level 2 data files. Positive values of precision in MLS products indicate that retrievals depend mainly on observed radiances rather than a priori estimates (precisions are

25 explicitly set negative by the software to flag retrievals that are significantly affected by their a priori estimates). The root-mean-square (RMS) precision of individual MLS temperature profiles over this region (see domain outlined in Fig. 1) during the four measurement campaigns was 0.5–1.3 K in v3 and 0.5–1.0 K in v4 for the 19 pressure levels included in this validation. The corresponding RMS precision of individual water vapour volume mixing ratio profiles was 4–39% in v3 and 4–8% in v4, and the RMS precision of individual ozone volume mixing ratio profiles between 10 and 261 hPa was 1–124% in v3 (100%

at 316 hPa) and 1–28% in v4 (490% at 316 hPa). In most cases, MLS precisions are fairly constant in mixing ratio space. Fractional precisions will therefore vary substantially for species with abundances that cover a large range (including water vapour and ozone).

Figure 2 shows mean profiles of temperature, water vapour and ozone from v3 and v4 within 1000 km of one of more of the launch sites (circular shaded areas in Fig. 1). Differences in mean profiles of temperature, water vapour and ozone based



on these two versions of MLS data are small, particularly at stratospheric pressure levels (10–68 hPa). Mean temperatures in the upper troposphere (100–316 hPa) are colder in v4 than in v3 by 0.24–0.82 K (all differences are significant when both measurement and statistical uncertainty are accounted for). The mean v4 temperature profile is also colder than v3 at 31–38 hPa (by approximately 0.2 K) and warmer than v3 at 56 hPa ($0.23\pm0.13$ K) and 10 hPa ($0.16\pm0.12$ K). Mean water vapour mixing ratios in the upper troposphere (215–316 hPa) are smaller in v4 than in v3 (with a maximum relative bias of $-29\pm5\%$ at 316 hPa), but slightly larger in v4 than in v3 at 147 hPa ($11\pm3\%$). Differences in the remainder of the profile are within $\pm3\%$. The most significant change in ozone is a reduction in vertical gradients in the upper troposphere and lower tropopause layer (100–316 hPa) in v4 relative to v3. This vertical homogenization results in better qualitative agreement with the vertical structure of mean ozonesonde profiles from independent observations over Lhasa and Kunming during boreal summer (Bian et al., 2012), and includes decreases of approximately 5–16% in mean ozone mixing ratios in the tropopause layer (83–147 hPa) and in the lower part of the upper troposphere ($-10\pm3\%$ at 261 hPa and $-27\pm4\%$ at 316 hPa). The mean profiles shown in Fig. 2 are based on slightly different samples due to differences in the retrieval algorithm and quality control criteria. Specifically, v4 provides increased data yields in this region relative to v3 (10% more temperature profiles, 32% more water vapour profiles and 29% more ozone profiles). Relative differences between v3 and v4 are effectively unchanged when the comparison is limited to retrievals that meet quality control criteria in both v3 and v4.

## 2.3 Validation methodology

Differences between sonde measurements and MLS retrievals can arise from several factors, including differences in vertical resolution or interpolation techniques, measurement errors in the sonde and MLS profiles, spatiotemporal inhomogeneities due to synoptic variability, and smoothing associated with the horizontal extent of the MLS footprint. The estimated response time of the CFH and ECC instruments are both on the order of 10 s to one minute. At typical ascent rates of 5–7 m s$^{-1}$, this corresponds to a vertical resolution of 50–400 m. By contrast, the vertical resolution of MLS profiles is on the order of a few kilometres (3.6–5.0 km for temperature, 2.0–3.7 km for water vapour and $\sim$2.5 km for ozone). The radiosonde profiles must therefore be resampled to match the lower vertical resolution of the MLS profiles. Here, we resample the radiosonde profiles of temperature, water vapour and ozone by applying the MLS forward model smoothing operator and appropriate averaging kernels (Read et al., 2006; Livesey et al., 2013, 2015). This approach to resampling the sonde profiles at MLS resolution differs notably from the linear interpolation method used by Yan et al. (2015).

First, the resolution of the observed in situ profile is degraded to the resolution of the MLS product using the equation

$$\bar{X}_{\mathrm{s}} = X_{\mathrm{s}}\eta^{\mathrm{T}}\left(\eta\eta^{\mathrm{T}}\right)^{-1}, \tag{1}$$



where $X_s$ is the sonde profile at its original resolution, $\eta^T \left( \eta \eta^T \right)^{-1}$ is the forward model smoothing operator (with $\eta$ dependent on the sonde and MLS pressure profiles as described by Read et al., 2006) and $\bar{X}_s$ is the sonde profile sampled at MLS resolution. The reduced-resolution sonde profile is then convolved with the averaging kernel using the equation

$$\hat{X}_s = X_{ap} + \left[ \bar{X}_s - X_{ap} \right] \mathbf{A}, \tag{2}$$

where $X_{ap}$ is the a priori profile for collocated retrieval and $\mathbf{A}$ is the averaging kernel. The resulting profile $\hat{X}_s$ is appropriate for direct comparison with the collocated MLS profile. Forward smoothing and convolution of water vapour profiles are done using the logarithm of water vapour volume mixing ratio as recommended by Read et al. (2007), while forward smoothing and convolution of temperature and ozone profiles are done using temperature and ozone volume mixing ratio directly (see also Livesey et al., 2015, and references therein). MLS averaging kernels differ by variable and data version. Sonde profiles are

convolved with v3 averaging kernels for validating v3 retrievals and with v4 averaging kernels for validating v4 retrievals.

Appropriate collocation criteria for MLS validation may vary by region, season or variable of interest, and should be evaluated independently for each validation campaign. Vömel et al. (2007a) used CFH measurements to validate Aura MLS version 1 and 2 observations of water vapour and found that their results were largely insensitive to the choice of distance thresholds up to 900 km and time difference thresholds up to 12 h. We begin by choosing the geographically closest MLS retrievals of

temperature, water vapour and ozone within $\pm 6$ h of balloon launch that satisfy the quality control criteria outlined in Section 2.2. If any of these retrievals are within 1000 km of the launch site, then we choose the geographically closest retrieval for comparison. If not, then we extend the time window to $\pm 12$ h and repeat the process. This two-step selection process allows us to preferentially select retrievals from orbits that are close in time to the in situ measurements, limiting spurious effects caused by sampling different parts of the diurnal cycle (although sensitivity analysis indicates that these effects are small)

while maximizing the validation sample size. Our conclusions are qualitatively robust to reasonable changes in these criteria: sensitivity to choices of smaller distance or time thresholds is mainly limited to data yields, with no major changes in bias statistics. Application of the combined collocation and quality control criteria eliminates only two ozone profiles from the analysis, and those two profiles are eliminated only from the v3 validation (both profiles are successfully matched to valid v4 ozone retrievals). The vast majority of profiles are matched within $\pm 6$ h (76–85 %, depending on the variable and data version),

with a mean time difference for all matched profiles of approximately 3.5 h. Distances between the launch site and the nominal center of the matched MLS footprint range from 37 km to 983 km, with a mean of approximately 500 km.

We report temperature biases as absolute differences in Kelvins; however, we report biases in water vapour and ozone mixing ratios as relative differences. The use of relative differences for water vapour and ozone accounts for variations of 2–3 orders of magnitude in typical concentrations of these species within the upper troposphere and lower–middle stratosphere,

and facilitates comparison with previous validation studies. We include a brief summary of absolute ozone biases for context, as many of the sources of error for MLS ozone retrievals act on absolute mixing ratios rather than relative mixing ratios.





Relative differences are defined with respect to the sonde measurement, and are calculated using the equation

$$\delta(p) = \frac{X_{\mathrm{M}}(p) - \hat{X}_{\mathrm{s}}(p)}{\hat{X}_{\mathrm{s}}(p)}, \tag{3}$$

where $X_{\mathrm{M}}(p)$ is the MLS retrieval at a given pressure level and $\hat{X}_{\mathrm{s}}(p)$ is the sonde measurement convolved to that level using Eqns. 1–2. We report three bias statistics at each level for each variable: the arithmetic mean bias, the median bias and the root-mean-square (RMS) bias. The range of biases at each level is indicated by twice the standard error of the mean bias (an approximate 95% confidence interval around the mean) and the interquartile range (which spans the middle 50% of biases at each level). For water vapour and ozone, non-robust statistical measures (mean, standard error and RMS) are calculated using absolute differences and then normalized by the mean of the convolved sonde observations at each level.

## 3 Results

### 3.1 Temperature

Figure 3 shows temperature profiles and biases based on RS92 radiosonde measurements and MLS v3 and v4 retrievals collected near Lhasa on 18 May 2012. The balloon was launched 24 minutes prior to the MLS retrieval and had reached 191 hPa at the time of the MLS overpass. The center of the MLS footprint was located 242 km due east of the radiosonde launch site. Application of the forward smoothing function and the v3 or v4 averaging kernel eliminates much of the fine structure in the radiosonde profile, but the resulting low-resolution profiles are consistent with the vertical structure of the in situ measurements at kilometre scales. Both v3 and v4 MLS retrievals are colder than the RS92 measurements in the middle stratosphere (10–32 hPa) and in the lower stratosphere and tropopause layer (56–100 hPa), while the v4 retrieval is substantially colder than both RS92 measurements and the MLS v3 retrieval in the upper troposphere (215–316 hPa). Differences between v3 biases and v4 biases are due in part to differences in the MLS retrievals and in part to differences in the RS92 profile convolved to MLS pressure levels. Both of these factors potentially reflect changes in the averaging kernel and the a priori profile (see Section 2.3), while the former also reflects changes in how the retrieval algorithm processes the observed radiances. The following discussion is based on a statistical analysis of 82 profiles, including the profile shown in Fig. 3.

Figure 4 shows mean, median and RMS biases for MLS v3 and v4 temperature retrievals relative to RS80/RS92 temperature measurements. Bias statistics at each level are based on the subset of radiosondes that reach that level, with sample sizes that range from 43 measurements at 10 hPa to 82 measurements at 316 hPa. Mean and median temperature biases agree well at most levels, although cold biases near the tropopause (68–83 hPa) are slightly larger in the median than in the mean. Both v3 and v4 agree well with the sonde profiles in the same portions of the profile, with layers of near-zero mean biases at 18, 46 and 100–121 hPa alternating with layers of cold biases centered at 10–12 hPa ($-1.5\pm0.5$ K in v3; $-1.3\pm0.5$ K in v4), 26–32 hPa ($-1.5\pm0.3$ K in v3; $-1.7\pm0.3$ K in v4), 68–83 hPa ($-1.8\pm0.4$ K in both v3 and v4) and 147–261 hPa ($-2.3\pm0.3$ K in v3; $-2.6\pm0.4$ K in v4). Vertical oscillations in the bias profile are a well-known feature of previous versions of MLS temperature retrievals (Schwartz et al., 2008), although their underlying causes are still not well understood (Livesey et al., 2013).





The pressure-weighted mean temperature bias in the stratosphere (10–56 hPa) is $-0.9\pm0.2$ K for both v3 and v4. RMS biases in this layer are nearly constant at about 2 K, generally increasing with height, with slight reductions from v3 to v4. The magnitude of the pressure-weighted mean temperature bias in the tropopause layer (68–147 hPa) has increased slightly from $-0.6\pm0.3$ K in v3 to $-0.9\pm0.2$ K in v4, despite a slight reduction in RMS bias in this layer. The magnitude of the pressure-weighted mean temperature bias in the upper troposphere (178–261 hPa) has also increased slightly, from $-2.4\pm0.3$ K in v3 to $-3.0\pm0.4$ K in v4 (316 hPa is omitted from this layer average, as temperature retrievals at this layer are not recommended for scientific use). However, unlike the tropopause layer, RMS biases in the upper troposphere are larger in v4 than v3. This increase in RMS biases (and the similar increase in IQR) indicates increased noise in upper tropospheric temperature retrievals in v4 relative to v3.

In addition to RS80 and RS92 radiosondes, 18 InterMet (IMet) radiosondes were launched at Lhasa (7) and Linzhi (11). All but one of the IMet radiosondes were launched together with an RS92 radiosonde, allowing for a comparative evaluation of MLS temperature biases relative to the two sets of radiosonde profiles. Figure 5 shows mean and RMS temperature biases relative to collocated IMet and RS92 radiosondes, with sample sizes ranging from 10 samples (10–12 hPa) to 17 samples (32–316 hPa). The two bias estimates are qualitatively identical; the only notable difference is that the magnitudes of mean and RMS biases relative to IMet are slightly larger than the magnitudes of mean and RMS biases relative to RS92 in the tropopause layer (68–100 hPa) for both v3 and v4 (note that this difference is consistent with the ~0.5 K magnitude of previously-reported biases between IMet and RS92 temperature measurements over California, but is opposite in sign and less persistent in altitude; Hurst et al., 2011). The overall agreement between bias estimates relative to IMet and bias estimates relative to RS92 indicates that the temperature validation presented here is robust to the choice of radiosonde instrumentation.

## 3.2 Water vapour

Figure 6 shows profiles of water vapour volume mixing ratio from 18 May 2012 based on CFH measurements and MLS v3 and v4 retrievals collected near Lhasa on 18 May 2012. These water vapour profiles correspond to the temperature profiles shown in Fig. 3. Application of the forward smoothing function and averaging kernel again eliminate the fine-scale structure in the CFH profile. The resulting low-resolution profiles are consistent with the kilometre-scale vertical structure of the CFH profile in the stratosphere (18–83 hPa), but are unable to accurately capture several deep layers of relatively dry and moist air below the tropopause (100–261 hPa). Biases in MLS v3 and v4 are small (within approximately ±20%) between 18 and 100 hPa. The large moist bias (70–80%) centered at 147 hPa may reflect weather-related horizontal gradients in water vapour in the upper troposphere and tropopause layer, which may be associated with horizontal variations in convective activity (current or previous) or radiatively-driven ascent. This spatial variability should average out at larger sample sizes, but its impact will still be reflected in the spread around the mean and median biases at these levels. We discuss this topic in more detail in Section 4. The large dry biases at 261 and 316 hPa reflect the impact of a sharp increase in the CFH water vapour measurements at approximately 320 hPa (not shown) on the values convolved to these two levels using Eqs. 1 and 2.

Figure 7 shows mean, median and RMS biases for MLS v3 and v4 retrievals of water vapour volume mixing ratio relative to CFH measurements. The number of samples used to calculate these statistics varies from two samples at 18 hPa to 30





samples at 121–316 hPa. The number of samples is 10 or larger from 32–316 hPa. Mean biases are within ±20% at all levels above 100 hPa. A statistically significant dry bias is identified at 22–32 hPa (−9±4% in v3; −8±4% in v4) and a statistically significant moist bias is identified at 68–83 hPa (6±4% in v3; 12±5% in v4). Mean biases between 121 and 261 hPa are generally negative in both v3 and v4, with the exception of v3 at 215 hPa. Pressure-weighted mean biases in this layer are −23±15% in v3 and −32±11% in v4. Mean biases at 316 hPa are near zero in both v3 and v4. Substantial reductions in the RMS bias, standard error and IQR at 316 hPa in v4 relative to v3 indicate that retrieved values at this level are substantially less noisy in v4.

The pressure-weighted mean bias in the stratosphere (18–56 hPa) is −3±2% in v3 and −2±2% in v4, while the pressure-weighted mean bias in the tropopause layer (68–147 hPa) is −9±5% in v3 and −11±6% in v4. The magnitude of the pressure-weighted mean dry bias in the upper troposphere (178–316 hPa) increases from v3 (−13±18%) to v4 (−23±12%), consistent with the systematically drier upper troposphere in v4 relative to v3 (Fig. 2b). Median biases in water vapour volume mixing ratio are similar to mean biases at pressures less than or equal to 100 hPa, but diverge slightly from mean biases at several levels with pressures greater than 100 hPa, particularly for v4. Differences between mean and median biases in the upper troposphere indicate that the mean bias may slightly overestimate the dry bias in this layer. Approximately 70% of the observed biases between 121 and 261 hPa are negative for both v3 and v4, indicating that MLS water vapour retrievals in the upper troposphere over the eastern Tibetan Plateau are typically drier than collocated CFH measurements.

RMS biases in water vapour volume mixing ratio through most of the UTLS are similar between v3 and v4. Values of RMS bias are less than 20% between 18 and 83 hPa, increase gradually to approximately 50% at 147–215 hPa, and then increase sharply in v3 to approximately 100% at 261 hPa and 150% at 316 hPa, and to about 85% at both levels in v4. Slight increases in RMS bias through the tropopause layer and lower stratosphere (56–147 hPa) indicate that increased data yields in v4 (see Section 2.2) may also slightly increase overall noise levels in this vertical range.

## 3.3 Ozone

Figure 8 shows profiles of ozone from ECC measurements and v3 and v4 MLS retrievals collected near Lhasa on 18 May 2012. These ozone profiles correspond to the temperature profiles shown in Fig. 3 and the water vapour profiles shown in Fig. 6. The ozone profile based on ECC measurements convolved to MLS pressure levels using the v3 averaging kernels is almost identical to that convolved to MLS pressure levels using the v4 averaging kernels, and both reduced-resolution profiles are consistent with the vertical structure of the underlying in situ measurements. By contrast, the v3 and v4 retrievals differ substantially from each other between 68 and 316 hPa, with biases of up to ±40% relative to the ECC measurements that often change sign between v3 and v4. One of the goals of the MLS v4 development was to reduce the degree of unrealistic vertical structure reported in v3 UTLS ozone profiles. This was accomplished by splitting the retrieval of ozone away from that of other species (notably, carbon monoxide and nitric acid) and neglecting channels that the retrievals were unable to fit accurately.

Figure 9 shows mean, median and RMS biases for MLS v3 and v4 retrievals of ozone volume mixing ratio relative to ECC measurements. The number of samples used to calculate these statistics varies from a minimum of 38 samples at 10 hPa (37 for v3) to a maximum of 69 samples at 316 hPa (67 for v3). Mean and median biases in ozone volume mixing ratio are positive





or statistically indistinguishable from zero throughout the UTLS except for at 100 hPa (where mean biases are −30±13% in v3 and −21±9% in v4 and median biases are −20% in v3 and −18% in v4). Positive biases through most of the stratosphere (18–83 hPa) indicate that MLS v3 and v4 generally overestimate ozone concentrations in this region during boreal summer. The maximum biases are located at 83 hPa (72±11% in v3; 63±10% in v4). This bias profile differs from bias profiles generated

by comparisons with most ozonesonde profiles in this latitude range (Jiang et al., 2007) and by comparisons with other satellite datasets (Tegtmeier et al., 2013), and may be regionally and/or seasonally specific. In particular, the proximity of the balloon launch sites to the center of the boreal summer lower stratospheric 'ozone valley' (Zhou et al., 1995; Tobo et al., 2008) may contribute to the large positive biases at 83 hPa. We discuss this feature further in Section 4.

The pressure-weighted mean bias in the stratosphere (10–56 hPa) is approximately the same (13±1%) in v3 and v4, while

the pressure-weighted mean bias in the tropopause layer (68–147 hPa) has decreased substantially from v3 (27±5%) to v4 (12±4%). The pressure-weighted mean bias in the upper troposphere (178–261 hPa) has changed from slightly negative in v3 (−11±15%) to slightly positive in v4 (12±9%), and its vertical profile is much more consistent in v4 than in v3 (as with temperature, 316 hPa is omitted from this layer average as ozone retrievals at this level are not recommended for scientific use). Median biases are similar to mean biases through most of the UTLS, with the exception of levels affected by large negative

outliers (e.g., 12 hPa). The consistency between mean and median biases in the stratosphere indicates that high biases in MLS ozone retrievals between 18 and 83 hPa are robust to statistical assumptions: almost all (∼92%) of the calculated biases in this layer are positive.

The vertical structure of RMS biases in ozone mixing ratio largely mirrors the vertical structure of mean biases. The pressure-weighted mean RMS bias in the upper troposphere and tropopause layer (68–261 hPa) is smaller in v4 (58%) than in v3 (89%),

with particularly pronounced improvements below 100 hPa. Despite increased data yields (see Section 2.2), noise in upper tropospheric ozone retrievals appears to be reduced in v4 relative to v3. These improvements are accompanied by a much more consistent mean bias profile below 121 hPa, with mean biases of approximately +10% throughout the upper troposphere, including at 316 hPa. Although ozone retrievals at 316 hPa are still not recommended for scientific use (Livesey et al., 2015), our validation indicates that these retrievals are much improved in v4.

MLS ozone retrievals are performed with respect to volume mixing ratio (unlike water vapour retrievals, which are performed with respect to the logarithm of volume mixing ratio). We therefore include a statistical evaluation of absolute biases in MLS v3 and v4 retrievals of ozone volume mixing ratio (Fig. 10) for context. Like the relative bias profiles, the absolute bias profiles are dominated by the high biases in the stratosphere, although the largest absolute biases are located at higher altitudes (38 hPa and above) than the largest relative biases (83 hPa). Pressure-weighted mean biases in the stratosphere are 378±56 ppbv in v3

and 368±54 ppbv in v4, pressure-weighted mean biases in the tropopause layer are 53±9 ppbv in v3 and 35±8 ppbv in v4, and pressure-weighted mean biases in the upper troposphere are −8±11 ppbv in v3 and 8±7 ppbv in v4.





## 4   Discussion

Aura is a sun-synchronous satellite, so that MLS observes the validation domain in the early morning (~01:45 local time; descending passes) and early afternoon (~13:45 local time; ascending passes). Most of the retrievals selected for validation were daytime observations: approximately 65% of the temperature validation, 73% of the water vapour validation and 60% of the

ozone validation is based on MLS retrievals made during ascending passes. We find no statistically significant differences between mean biases calculated for ascending retrievals and mean biases calculated for descending retrievals, with the exception of v3 temperature at two levels in the upper troposphere. Cold biases in v3 temperature retrievals at 178 hPa are reduced during daytime ($-1.1\pm0.5$ K) relative to nighttime ($-2.4\pm0.7$). This difference in mean biases persists up to 83 hPa, although the uncertainty windows overlap for pressures less than 178 hPa. By contrast, cold biases at 261 hPa are enhanced during daytime

($-3.5\pm0.9$ K) relative to nighttime ($-1.8\pm0.6$). Cold biases at 316 hPa are also enhanced for daytime retrievals ($-4.3\pm1.5$ K) relative to nighttime retrievals ($-0.2\pm1.2$ K), but these data are not recommended for use in scientific studies. Convective activity over the southeastern Tibetan Plateau peaks in the late afternoon (Fujinami et al., 2005), so that these differences may be caused by convective activity or cloud contamination that is undetected by the quality control criteria. However, they may also be attributable to systematic differences between the conditions over Linzhi in 2014 (where most of the sondes were launched

at times corresponding to descending passes) and the other measurement sites in 2010–2012 (where most of the sondes were launched at times corresponding to ascending passes). Qualitatively similar differences are identified in the v4 temperature validation, but these difference are not statistically significant.

The standard deviation, interquartile range and RMS of temperature, water vapour and ozone biases generally increase with increasing pressure, indicating that the spread in the calculated biases is greatest in the upper troposphere. For example,

standard deviations in 261 hPa temperatures based on radiosonde observations range from 0.8 K (Tengchong) to 2.0 K̇ (Naqu), whereas standard deviations in 261 hPa temperatures based on retrievals within 200 km of the measurement sites during the measurement campaigns are much larger, ranging from 2.5 K (Naqu) to 5.5 K (Lhasa) in v3 and from 3.0 K (Naqu) to 4.7 K (Tengchong) in v4. Further analysis of our validation results shows strong correlations between the magnitude of the calculated bias and the value retrieved by MLS in the upper troposphere and tropopause layer, particularly for temperature and ozone

(not shown). These correlations are uniformly positive, indicating that the variance in the collocated MLS retrievals is larger than the variance in the in situ measurements (i.e., that the in situ measurements are more tightly clustered around the mean value). The correlation coefficients between temperature retrievals and biases and between ozone retrievals and biases increase with increasing pressure, indicating that larger variance in the MLS retrievals is the primary source of the larger spread in the calculated biases. The primary implication is that MLS-based estimates of temperature, water vapour and ozone in the upper

troposphere are more reliable at monthly and seasonal time scales than at event time scales, where excessive noise can result in biases at several layers within the UTLS.

Excess variance in the MLS retrievals relative to the in situ measurements can arise from several sources, and is not necessarily spurious. These sources include increasing uncertainty and noise (due to greater attenuation of the radiance signals used in the MLS retrievals and potential contamination by ice particles in the upper troposphere), but also the increasing influence of




spatiotemporal variability. Spatiotemporal variability in the composition and thermodynamic structure of the upper troposphere may be related to variations in deep convective activity, wave activity, the location of the upper tropospheric anticyclone and other meteorological and climatological features. Many of these variations can be considered approximately random from the perspective of regular sampling at a fixed location, with effects that will be reflected in the variance of the MLS measurements

and the spread around the mean bias but will be negligible with respect to the mean itself. However, variations associated with seasonal climatological features (such as the mean position of the upper tropospheric anticyclone) or features that are unrepresented in the in situ data (such as active convection, which precludes radiosonde launch) may manifest as systematic biases.

The UTLS over the summertime Asian monsoon contains sharp horizontal gradients in temperature, water vapour and

10 ozone (Randel and Park, 2006; Park et al., 2007). Preferential sampling of collocated MLS retrievals upgradient of the measurement sites would result in an apparent high bias, while preferential sampling of retrievals downgradient of the measurement sites would result in an apparent low bias. For example, the in situ measurements used to validate MLS retrievals in this study were collected near the 'ozone valley' that develops in the UTLS during the Asian summer monsoon (Zhou et al., 1995; Tobo et al., 2008). This regional-scale ozone minimum is a result of repeated injection of low-ozone air by monsoon convection

(potentially augmented by chemical depletion in the tropopause layer), and propagates upward as part of the large-scale ascent associated with the monsoon anticyclone.

Figure 11 shows time-mean gridded spatial distributions of ozone at 83 hPa based on MLS v4 retrievals during the four measurement campaigns. The time-mean location and magnitude of the ozone minimum varied substantially among the measurement campaigns, but the local minimum was consistently located nearby to the measurement sites. Slightly more than a

20 third of the MLS retrievals collocated with our ozonesonde profiles were based on observation at locations with similar values to the measurement sites, but more of the retrievals were located upgradient (43% at locations with higher time-mean ozone concentrations) than were located downgradient (22% at locations with lower time-mean ozone concentrations). The mean difference between the time-mean values at the upgradient sites and the time-mean values at the launch sites (+26%) was also more than double the mean difference between the time-mean values at the downgradient sites and the time-mean values at the

25 launch sites (−12%). These results suggest that some portion of the high bias in MLS ozone at 83 hPa may be due to spatial sampling biases.

To more fully evaluate the possibility that preferential spatial sampling produces a high bias in MLS ozone at 83 hPa, we interpolate time-mean gridded MLS profiles to each measurement site using bilinear interpolation (the results are virtually identical when higher-order interpolation schemes are used) and compare these profiles with mean in situ observations collected

during the associated measurement campaign. The results are shown in Fig. 12. The shape of the bias profile is qualitatively robust. Moreover, the mean bias at 83 hPa appears to accurately capture the local biases at Tengchong and Linzhi but underestimate the local biases at Naqu and Lhasa. Weighting the results by the number of profiles at each site, the mean interpolated bias is 87% for v3 and 82% for v4. These values are larger than the mean biases calculated in Section 3.3 (70±11% for v3 and 66±10% for v4). We therefore find little evidence that systematic errors arising from preferential spatial sampling could

cause us to overestimate the high bias in MLS ozone mixing ratios at 83 hPa. A more plausible explanation may be propagation





of information from the a priori profile into the retrieval, in particular through the smoothing constraints in the MLS retrieval algorithms, which favor profiles whose shape (characterized by the vertical second derivative) is closer to that of the a priori. The MLS a priori profiles (which are taken from monthly zonal mean model output) begin to increase at a lower altitude with a more gradual vertical gradient in the tropopause layer than is typically observed over this region during the monsoon (Fig. 13;

see also Bian et al., 2012; Yan et al., 2015). This hypothesis is provisionally supported by a significant reduction in biases at 83 hPa calculated using observations made at Lhasa and Linzhi before monsoon onset ($44\pm14\%$ in v3; $26\pm15\%$ in v4) relative to during the monsoon ($82\pm15\%$ in v3; $78\pm12\%$ in v4). Sensitivity testing has shown that MLS retrievals are largely insensitive to constant offsets in the a priori profiles; however, the impacts of shifting tropopause-related gradients and other sharp features vertically within the a priori profiles have not yet been examined.

Temperature within the Asian monsoon anticyclone is warm relative to the zonal mean in the upper troposphere and cold relative to the zonal mean in the tropopause layer (Park et al., 2007). A broad temperature maximum was located to the west of the measurement sites in the upper troposphere (178–261 hPa), but time-mean temperature contours above the measurement sites were approximately zonal at 147 hPa and above. Despite some regional differences, biases between time-mean temperature profiles interpolated to each measurement site and mean temperature profiles based on radiosonde observations at the

corresponding measurement site (not shown) are comparable in both structure and magnitude to the mean and median biases shown in Fig. 4. The vertical structure of temperature biases before monsoon onset is similar to that during the monsoon; however, cold biases in the stratosphere are slightly enhanced (by $\sim$0.5–1 K) before monsoon onset relative to during the monsoon, while cold biases in the upper troposphere are slightly enhanced (by $\sim$1–1.5 K) during the monsoon relative to before monsoon onset. At 100–121 hPa, cold biases before the monsoon ($-0.8\pm0.6$ K in v3; $-1.3\pm0.5$ K in v4) are replaced by warm biases

during the monsoon ($0.9\pm0.6$ K in v3; $0.7\pm0.6$ K in v4). These quantitative changes in temperature bias oppose the changes in temperature structure that accompany monsoon onset (warming in the upper troposphere and cooling near the tropopause), and indicate that MLS underestimates seasonal changes in UTLS temperature associated with the establishment of the Asian monsoon anticyclone.

    The measurement sites are also located in the vicinity of sharp gradients in water vapour, with a broad maximum in the upper

troposphere and tropopause layer that transitions to an approximately south–north gradient in the stratosphere. The maximum in the upper troposphere is generally centered over the Bay of Bengal, south of the measurement sites, while the maximum in the tropopause layer is centered over the south slope of the Tibetan Plateau, almost directly above the measurement sites. Comparison of mean CFH profiles and time-mean MLS profiles interpolated to each measurement site (not shown) reveals regional variability, but no systematic differences relative to the mean bias profile shown in Fig. 7. The combined regional

biases are similar to the mean and median bias profiles discussed in Section 3.2. With the exception of 316 hPa in v3, standard deviations in MLS retrievals within 200 km of the measurement sites are comparable to (and sometimes even smaller than) standard deviations in the CFH measurements convolved to MLS pressure levels. Combined with the lack of strong correlations between water vapour retrievals and biases, this quantitative similarity indicates that the increase in the spread of water vapour biases with increasing pressure may be attributable to real variability with time and/or space scales similar to or smaller than the

collocation criteria, rather than noise in the retrievals. We conclude that MLS provides a reliable representation of water vapour



mixing ratios in the UTLS with respect to both mean values (mean biases within ±20%) and variance (although MLS v4 may underestimate the real variability of water vapour in the upper troposphere between 215–261 hPa, where standard deviations in MLS retrievals are less than 50% of the standard deviations in the CFH measurements). Retrievals of water vapour mixing ratios at 316 hPa are generally improved in v4 relative to v3, particularly in terms of variance (which is comparable to CFH-

5 derived variance in v4, but approximately double CFH-derived variance in v3). We find no significant differences in water vapour biases before monsoon onset relative to during the monsoon.

MLS v3 temperature and ozone retrievals at 316 hPa are not recommended for scientific use due to excessive noise, large biases and insufficient validation (Livesey et al., 2013, 2015); we now revisit these recommendations in the context of our results. The results of our validation analysis (Fig. 4) show a slight decrease in the mean ($-2.9\pm1.1$ K to $-2.7\pm1.5$ K) and

10 median ($-2.8$ K to $-1.8$ K) temperature biases at 316 hPa in v4 relative to v3, although these changes are not statistically significant. By contrast, there are strong indications that v4 temperature retrievals are noisier than v3 temperature retrievals at this level, as indicated by increases in the RMS bias (from 5.9 K to 7.5 K), the standard error of the mean bias (from 0.6 to 0.8 K) and the extent of the interquartile range around the median bias (from 6.2 K to 10.5 K). Users of v4 should continue to avoid the use of temperature retrievals at 316 hPa in scientific studies. Moreover, enhanced noise in v4 temperature retrievals

(relative to v3) extends upward to 178 hPa, indicating that users of MLS retrievals of upper tropospheric temperature should exercise care before using v4 (especially for studies of specific events). By contrast, our validation of MLS ozone retrievals (Fig. 9) shows a sharp reduction in the mean bias of ozone volume mixing ratio at 316 hPa in v4 ($10\pm30\%$) relative to v3 ($67\pm47\%$), along with reductions in RMS (from 204% in v3 to 128% in v4) and median (from 82% in v3 to 29% in v4) biases. Although ozone retrievals at this level remain noisy and additional evaluation is still needed, our results indicate that v4

represents a substantial improvement in ozone retrievals at 316 hPa (and throughout the upper troposphere) relative to v3.

Yan et al. (2015) presented a preliminary validation of MLS v2 and v3 water vapour and ozone retrievals using many of the soundings collected at Tengchong, Naqu and Lhasa. Our methodology differs from theirs in several respects, most notably in the approach used for convolving the sonde profiles to MLS levels (where we use the MLS averaging kernels and forward smoothing function as opposed to linear interpolation) and in the criteria used to select coincident MLS retrievals for validation.

Moreover, the inclusion of additional soundings (particularly those collected at around midnight local time over Linzhi) substantially reduces uncertainty windows around the mean and median biases and allows for a more comprehensive validation of retrievals collected during both ascending and descending satellite overpasses. The key features of the water vapour and ozone bias profiles are robust despite these differences in methodology, particularly the high biases in lower stratospheric ozone.

## 5   Summary and outlook

Aura MLS v3 and v4 retrievals of temperature, water vapour and ozone provide valuable information about the thermal structure and composition of the upper troposphere and stratosphere in the Asian monsoon anticyclone. We have presented a validation of these data in the UTLS (10–316 hPa) using in situ measurements collected using balloon-borne instruments over the Tibetan Plateau (Naqu, Lhasa and Linzhi) and adjacent regions (Tengchong, Yunnan) during four recent summers.



Temperature biases are largely similar between v3 and v4, with slightly smaller cold biases in v4 in the tropopause layer (68–147 hPa) and lower–middle stratosphere (10–56 hPa), but slightly larger cold biases in v4 in the upper troposphere (178–261 hPa). Vertical oscillations in the temperature bias profile that have existed since the initial public release (Schwartz et al., 2008) persist in v4. Retrievals at 316 hPa remain unsuitable for use in scientific studies, while increased variance in v4 through-

out the upper troposphere (178–261 hPa) may create issues for studies focused on individual events or using small sample sizes. Variances in MLS retrievals are several Kelvin larger than variances derived from radiosonde profiles, especially in the upper troposphere, reflecting the effects of noise on the retrievals. The upper troposphere over the validation domain is systematically colder by 0.2–0.8 K in v4 relative to v3, while changes to the retrieval algorithm and quality control criteria increase the data yield in this region by about 10%.

Biases in v3 and v4 water vapour retrievals in the stratosphere are also largely similar to each other, with a slightly smaller dry bias near 22–26 hPa and a slightly larger moist bias near 68–83 hPa. The vertical profile of mean biases between 121 and 261 hPa is more homogeneous in v4 than in v3, but at the cost of larger dry biases in v4 at 215 hPa. MLS v4 retrievals of water vapour in the upper troposphere are about 30% drier than collocated CFH measurements. This dry bias is more vertically homogeneous than previous estimates from other regions based on comparison with CFH measurements (which found mean

moist biases at 261 hPa Read et al., 2007; Vömel et al., 2007a) and considerably larger than estimates based on comparison with other satellite retrievals in the tropical or extratropical mean (Hegglin et al., 2013). RMS biases and other variance estimates are slightly larger between 56 and 178 hPa, perhaps due to large increases (∼32%) in data yield from v3 to v4. By contrast, variance is substantially reduced at 261 and 316 hPa, and is now largely consistent with estimates of variance derived from the CFH measurements.

Ozone retrievals are substantially improved in v4 relative to v3, particularly in the upper troposphere and tropopause layer: biases in ozone retrievals in the tropopause layer are significantly smaller, variance is reduced below 68 hPa (along with a 29% increase in data yield), and sharp gradients in the vertical profile of ozone biases in the upper troposphere are largely eliminated. The most influential change in ozone is a reduction of the vertical gradient of ozone mixing ratio between 100 and 316 hPa, which includes decreases in ozone mixing ratios in the tropopause layer (83–147 hPa) and in the lower part

of the upper troposphere (261–316 hPa). Despite these improvements, MLS ozone retrievals are biased high relative to ECC measurements through most of the stratosphere (18–83 hPa) and biased low relative to ECC measurements at 100 hPa. The bias profile contains a pronounced peak of about +70% at 83 hPa, which is not seen in biases relative to measurements made at most other ozonesonde sites (Jiang et al., 2007) or retrievals made by other satellites (Tegtmeier et al., 2013), and may therefore be specific to ozone retrievals in the vicinity of the Asian monsoon anticyclone. Detailed analysis indicates that this

bias is unlikely to result from preferential sampling of higher ozone mixing ratios upgradient from the nearby 'ozone valley'. We propose that this persistent high bias may instead propagate into the retrieval via smoothing towards the a priori profile, which does not adequately represent the very sharp vertical gradient in ozone concentrations near the tropopause over this region.

Overall, our validation indicates that v4 represents an improvement on v3. This improvement is particularly apparent for

ozone, but is also manifest in increased data yields and small improvements in the bias profiles for temperature (at 147 hPa and





above) and water vapour. Temperature retrievals in the upper troposphere (178–261 hPa) are more problematic, as v4 shows a larger cold bias and larger variance than v3 at these levels. Several features of this validation differ from previously published estimates of global and tropical biases in MLS retrievals, including the structure and magnitude of high biases in ozone through much of the stratosphere (which are substantially larger than previous estimates, particularly at 68–83 hPa Jiang et al., 2007;

Tegtmeier et al., 2013) and the magnitude of dry biases in the upper troposphere (which are slightly larger and more vertically homogeneous than previous estimates; Read et al., 2007; Hegglin et al., 2013). These results will help to facilitate future studies of the thermal structure and composition of the UTLS in the Asian monsoon anticyclone, and will contribute to future improvements in the MLS retrieval algorithm and data products in this critical region of the atmosphere.

**Appendix A: Quality control criteria**

In addition to the collocation criteria (within 1000 km and ±12 h), we have selected only high-quality MLS retrievals for validation. Livesey et al. (2013) and Livesey et al. (2015) recommended quality control criteria for v3 and v4, respectively. We slightly modify their recommendations to ensure that all profiles selected for comparison are valid throughout the 316–10 hPa vertical range. These modifications result in more restrictive criteria, at the potential cost of selecting retrievals that are farther from the balloonsonde launch site than the closest viable retrieval at some levels. The quality control criteria we use

are reproduced below for convenience; readers requiring further details should refer to Livesey et al. (2013) or Livesey et al. (2015).

### A1   Temperature

For v3 temperature, the `Convergence` flag must be less than 1.2; the `Quality` flag must be greater than 0.65; the `Status` flag must be even; the fifth ("low cloud") bit of the `Status` flag must not be set for either of the following two retrievals in

the orbit; and `L2gpPrecision` must be positive at all levels between 316 and 10 hPa.

For v4 temperature, the `Convergence` flag must be less than 1.03; the `Quality` flag must be greater than 0.9; the `Status` flag must be even; the MLS-retrieved ice water content (IWC) at 215 hPa must be less than $0.005\,\mathrm{mg\,m^{-3}}$; and `L2gpPrecision` must be positive at all levels between 316 and 10 hPa, and must be less than or equal to 0.7 at 261 hPa and 0.825 at 215 hPa.

### A2   Water vapour

For v3 water vapour, the `Convergence` flag must be less than 2.0; the `Quality` flag must be greater than 1.3; the `Status` flag must be even; the fourth ("high cloud") and fifth ("low cloud") bits of the `Status` flag must not be set; and `L2gpPrecision` must be positive at all levels between 316 and 10 hPa.

For v4 water vapour, the `Convergence` flag must be less than 2.0; the `Quality` flag must be greater than 1.45; the

`Status` flag must be even; and `L2gpPrecision` must be positive at all levels between 316 and 10 hPa.





## A3 Ozone

For v3 ozone, the `Convergence` flag must be less than 1.18; the `Quality` flag must be greater than 0.6; the `Status` flag must be even; `L2gpPrecision` must be positive at all levels between 316 and 10 hPa; and `L2gpValue` must be greater than $-0.3 \times 10^{-6}$ at 316 hPa and greater than $-0.15 \times 10^{-6}$ at all other levels. Occasional negative values in the ozone retrievals are caused by low signal-to-noise ratios (likely due to low ozone mixing ratios in the troposphere). The inclusion of these negative values is necessary to avoid high biases in measures of the statistical center and low biases in measures of statistical spread (Livesey et al., 2015).

For v4 ozone, the `Convergence` flag must be less than 1.03; the `Quality` flag must be greater than 1.0; the `Status` flag must be even; and `L2gpPrecision` must be positive at all levels between 316 and 10 hPa.

*Acknowledgements.* Support for the balloon soundings at Tengchong was provided by the National Natural Science Foundation of China under grant 40875014. Support for the balloon soundings at Naqu, Lhasa and Linzhi was provided the Special Fund for Meteorological Research in the Public Interest under grant GYHY201106023 and the Science and Technological Innovation Team Project of Chinese Academy of Meteorological Science under grants 2011Z003 and 2013Z005. The validation analysis was supported by a Young Thousand Talents fellowship at Tsinghua University. The measurement campaigns were supported by the Tengchong Meteorological Bureau in Yunnan and the Naqu Meteorological Bureau, Lhasa Meteorological Bureau and Linzhi Meteorological Bureau in Tibet. Work at the Jet Propulsion Laboratory, California Institute of Technology was performed under a contract with the National Aeronautics and Space Administration. MLS data were obtained from the Atmospheric Composition Data and Information Services Center FTP archive (ftp://acdisc.gsfc.nasa.gov) hosted by NASA Goddard Space Flight Center. We thank Michelle Santee, Irina Gerasimov and James E. Johnson for facilitating early access to the MLS version 4 data; Yonghong Ma and Yong Zhang from the Tibet Meteorological Bureau; Weiguo Wang from Yunnan University and the staff members of Kunming Observatory; Kejia Jia at the Linzhi Meteorological Bureau; and Wei Li, Jianyang Song, Jin Ma and Shumeng Sun from the Chinese Academy of Meteorological Sciences.





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





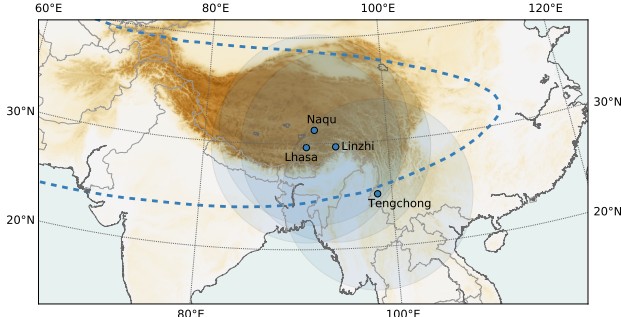

**Figure 1.** Locations of the radiosonde launch sites. The circular shaded areas cover a 1000 km radius from each site. The dashed contour corresponds to the 16 770 gpm contour of 100 hPa geopotential height from the ERA-Interim reanalysis, averaged over July and August 1979–2014. The shading indicates topography, with darker colors corresponding to higher surface altitudes.

**Table 1.** Radiosonde site information for data used in this validation. The rightmost column lists the number of profiles for which a collocated Aura MLS profile was identified (see Section 2.3)

| Site | Geolocation | Altitude | Period | Number of profiles | | |
|------|-------------|----------|--------|-------|--------|--------|
| | | | | $T^a$ | $H_2O$ | $O_3{}^b$ |
| Tengchong, Yunnan | 25.00°N, 98.50°E | 1656 m | August 2010 | 10 | 10 | 10 |
| Naqu, Tibet | 31.29°N, 92.04°E | 4500 m | August 2011 | 13 | 2 | 4 |
| Lhasa, Tibet | 29.66°N, 91.14°E | 3650 m | May–July 2012 | 28 | 8 | 25 |
| Linzhi, Tibet | 29.67°N, 94.33°E | 2992 m | June–July 2014 | 31 | 10 | 30 |

[a] Listed quantities are temperature profiles collected using Vaisala RS80 and RS92 instruments. Eighteen temperature profiles were also collected using InterMet (IMet) instruments at Lhasa (7) and Linzhi (11).

[b] Two $O_3$ profiles collected at Tengchong (17 and 22 August 2010) could not be matched to v3 MLS ozone retrievals that meet the quality control and collocation criteria. Both profiles were successfully matched to v4 MLS retrievals and included in the validation of v4 ozone.

Wright, J. S., Fu, R., Fueglistaler, S., Liu, Y. S., and Zhang, Y.: The influence of summertime convection over Southeast Asia on water vapor in the tropical stratosphere, J. Geophys. Res. Atmos., 116, doi:10.1029/2010JD015416, d12302, 2011.

Xu, X., Lu, C., Shi, X., and Gao, S.: World water tower: An atmospheric perspective, Geophys. Res. Lett., 35, doi:10.1029/ 2008GL03586, l20815, 2008.

5 Yan, X., Zheng, X., Zhou, X., Vömel, H., Song, J., Li, W., Ma, Y., and Zhang, Y.: Validation of Aura Microwave Limb Sounder water vapor and ozone profiles over the Tibetan Plateau and its adjacent region during boreal summer, Science China Earth Sciences, 58, 589–603, doi:10.1007/s11430-014-5014-1, 2015.

Zhou, X. J., Luo, C., Li, W. L., and Shi, J. E.: Variations of total ozone amount in China and the ozone low center over Tibetan Plateau, China Sci. Bull., 40, 1396–1398, (in Chinese), 1995.





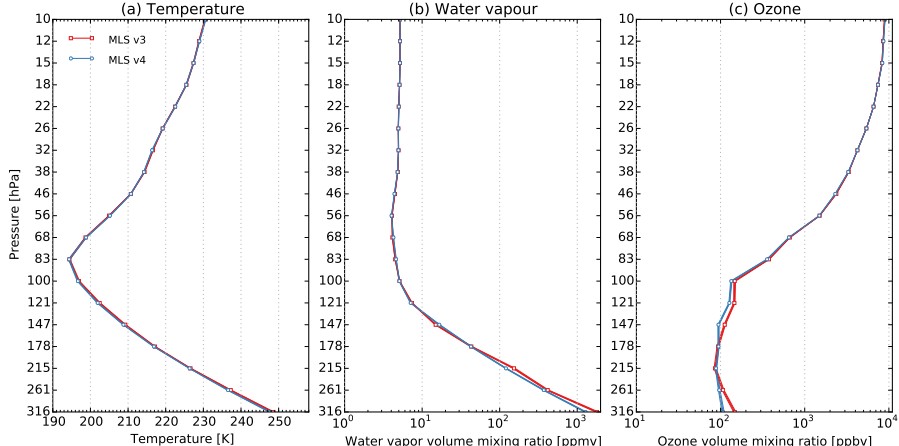

**Figure 2.** Mean profiles of (a) temperature, (b) water vapour volume mixing ratio and (c) ozone volume mixing ratio from MLS v3 and MLS v4 within one or more of the circular shaded areas outlined in Fig. 1 during the months corresponding to the four measurement campaigns (August 2010, August 2011, May–July 2012 and June–July 2014). Uncertainty bounds represent combined measurement (estimated retrieval precision) and statistical (twice the standard error of the mean) uncertainties.

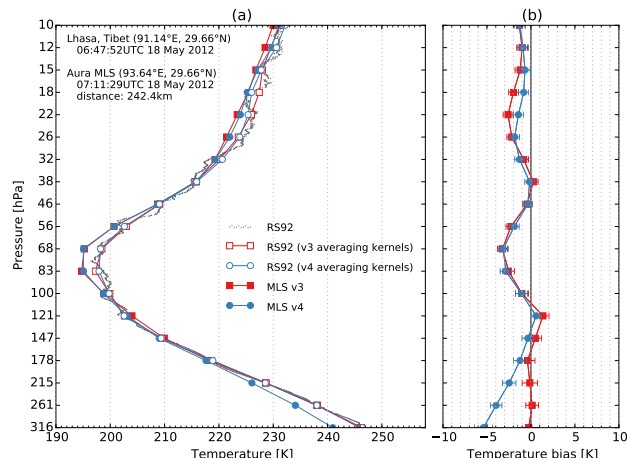

**Figure 3.** (a) RS92 and MLS v3 and v4 temperature profiles measured at Lhasa, Tibet on 18 May 2012. The satellite overpass was offset from the launch site by 242 km (0° latitude, 2.5° longitude). The balloon was launched 24 minutes prior to the overpass and reached 191 hPa at the time of the overpass. (b) Absolute biases (in K) between the MLS v3/v4 profiles and the sonde profile interpolated to MLS pressure levels using the v3/v4 MLS forward functions and averaging kernels. Error bars represent the MLS measurement uncertainty.





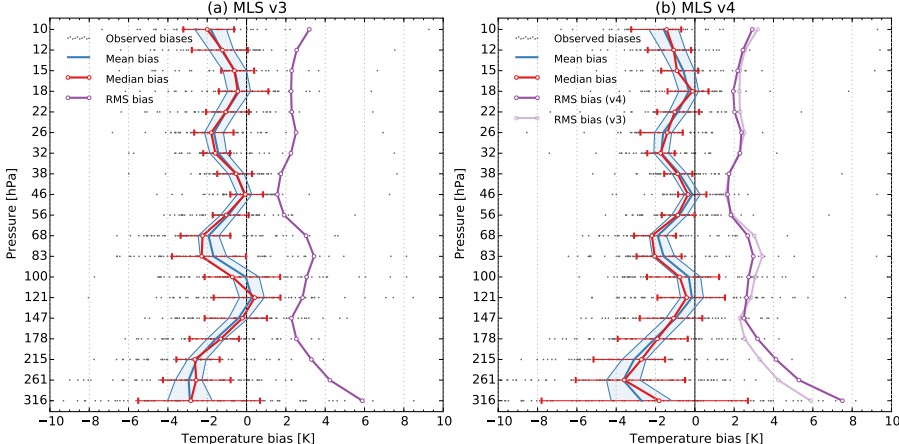

**Figure 4.** Mean, median and root-mean-square biases for (a) MLS v3 and (b) MLS v4 temperature profiles relative to RS80 and RS92 temperature profiles interpolated to MLS pressure levels using the respective MLS forward function and averaging kernel. Uncertainty bounds on the mean bias represent twice the standard error of the mean, while error bars on the median bias indicate the inter-quartile range (IQR). Results for individual profiles are shown as grey points.

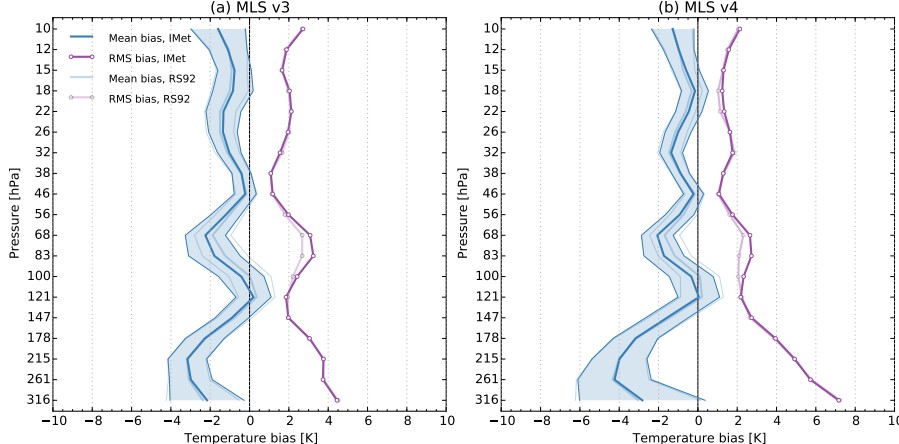

**Figure 5.** Mean and root-mean-square biases for (a) MLS v3 and (b) MLS v4 temperature retrievals relative to 17 temperature profiles at Lhasa and Linzhi measured using both InterMet (IMet) and RS92 radiosondes. Uncertainty bounds on the mean bias represent twice the standard error of the mean.





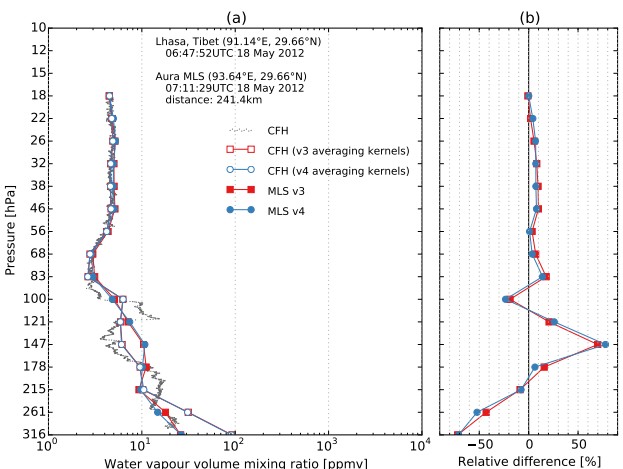

**Figure 6.** As in Fig. 3, but for CFH, MLS v3 and MLS v4 profiles of water vapour volume mixing ratio.

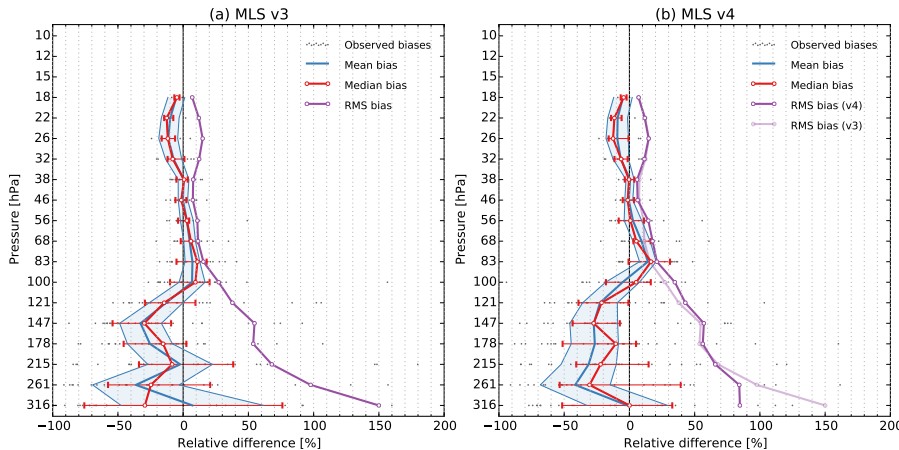

**Figure 7.** As in Fig. 4, but for relative biases between CFH and (a) MLS v3 and (b) MLS v4 profiles of water vapour volume mixing ratio. The mean and RMS biases (and associated uncertainties) are calculated from absolute differences and then normalized relative to the mean CFH-derived mixing ratio at each level. The median bias and IQR are calculated using relative differences from each validation profile.





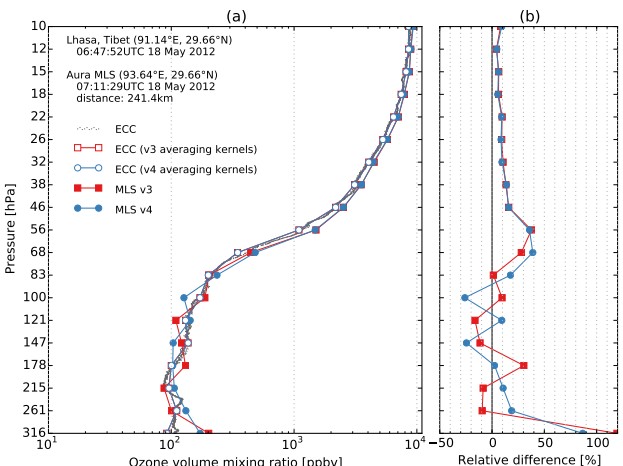

**Figure 8.** As in Fig. 3, but for ECC, MLS v3 and MLS v4 profiles of ozone volume mixing ratio.

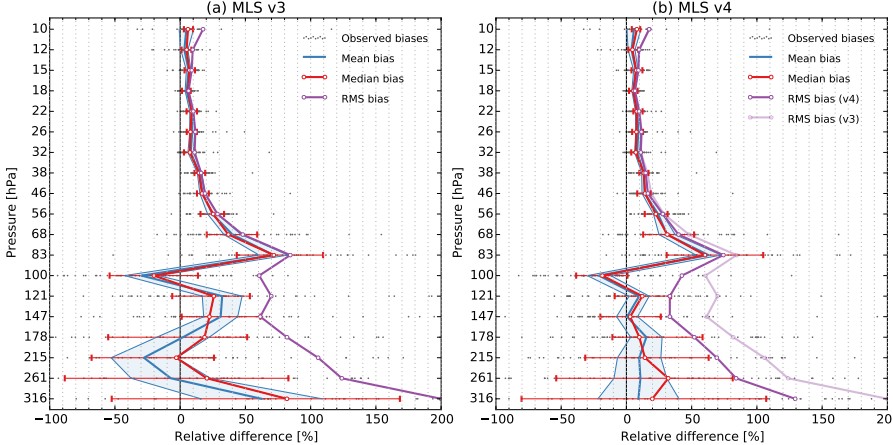

**Figure 9.** As in Fig. 4, but for relative biases between ECC and (a) MLS v3 and (b) MLS v4 profiles of ozone volume mixing ratio.





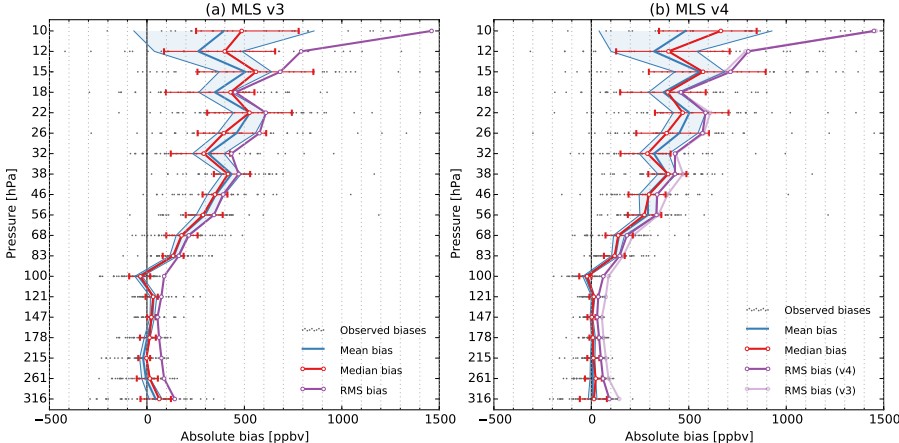

**Figure 10.** As in Fig. 4, but for absolute biases between ECC and (a) MLS v3 and (b) MLS v4 profiles of ozone volume mixing ratio.

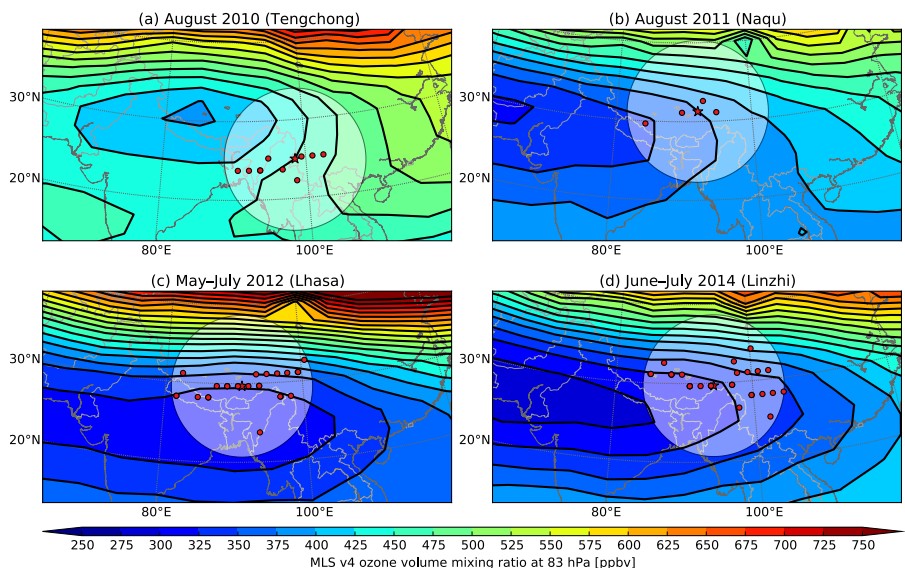

**Figure 11.** Balloon launch locations (red stars), centers of footprints of MLS observations collocated to balloons launched at each location (red circles) and mean spatial distributions of ozone volume mixing ratio at the 83 hPa level during the measurement campaigns at (a) Tengchong, Yunnan in August 2010, (b) Naqu, Tibet in August 2011, (c) Lhasa, Tibet in May–July 2012 and (d) Linzhi, Tibet in June–July 2014. The gridded values are inverse-distance weighted averages on a 5° longitude by 2.5° latitude grid, with weighted contributions from all valid measurements within 10° longitude and 10° latitude of the grid cell center.





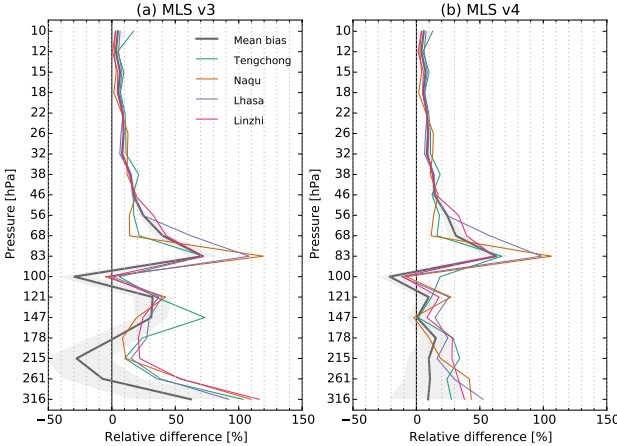

**Figure 12.** Relative differences between ECC ozonesonde observations and retrievals of ozone volume mixing ratio from (a) MLS v3 and (b) MLS v4. Grey lines and shading represent the mean bias and twice the standard error of the mean bias from the core validation analysis, as shown in Fig. 9. Colored lines represent relative differences between mean ozonesonde observations convolved to MLS pressure levels and time-mean gridded MLS observations during each study period (see, e.g., Fig. 11) interpolated to the respective ozonesonde launch site.

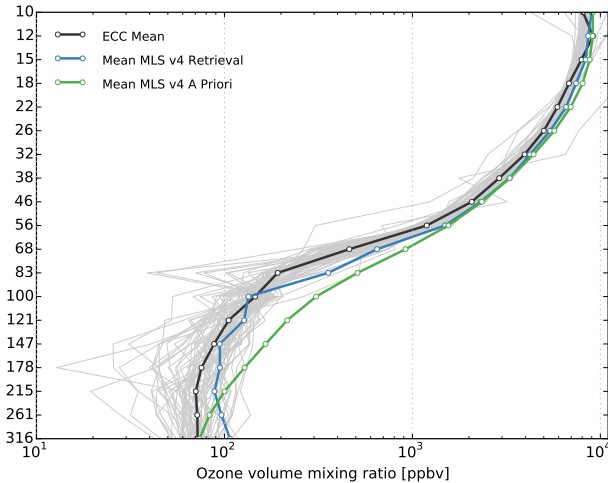

**Figure 13.** Vertical profiles of ozone mixing ratio from ECC profiles convolved to MLS pressure levels (mean in dark grey; individual profiles in light grey), MLS v4 retrievals in the validation domain (blue), and MLS v4 a priori profiles in the validation domain (green).