# Peer review of "Validation of Aura MLS retrievals of temperature, water vapour and ozone in the upper troposphere and lower-middle stratosphere over the Tibetan Plateau during boreal summer"

_Atmospheric Measurement Techniques, 2015_

## Referee Comment (RC1) · Anonymous Referee #1 · 15 Feb 2016

This is a very informative and thorough paper describing the latest version of MLS temperature, water vapor and ozone retrievals (v4) as well as the implications of the changes on scientific questions such as the composition of the UTLS over South East Asia during the summer monsoon season. After reading the manuscript, a few questions came to mind: (1) In section 2.2, the authors note ".. v4 provides increased data yields in this region relative to v3..." It's not entirely clear as to why this is the case. Were flag thresholds revised, for instance, and if so, why? (2) What is the impact of clouds in v4? Are clouds being addressed differently in this new version? Some suggestions for a few figures: (Fig 2) It is very difficult to see the uncertainties in these

plots as presented, since the magnitudes of the mean values are much larger and hence mask the magnitudes of the uncertainties. Consider plotting the uncertainties separately or instead of the means, if that is the focus of the plot. In addition, why average together different years and different seasons into one plot? (Fig 4) It is difficult to see if any differences exist between the error bars on the mean (set as 2-SEM) and the error bars on the median (set as the IQR) in both (a) and (b). Consider offsetting them in the vertical. (Figs 7, 9, 10, 12) The "x" axes in these figures should be labelled with the tracer names, similar to how it was done in Figs 5 and 6, for quick visual reference.

---

## Author Comment (AC1) · 20 Apr 2016

Our thanks to Anonymous Referee #1 for their contributions, and particularly for raising several pertinent questions that have helped to improve the clarity and completeness of the manuscript. Our responses and a summary of related changes to the manuscript are given below.

**(1) In section 2.2, the authors note "... v4 provides increased data yields in this region relative to v3..." It's not entirely clear as to why this is the case. Were flag thresholds revised, for instance, and if so, why?**

**(2) What is the impact of clouds in v4? Are clouds being addressed differently in this new version?**

The answers to these two questions are coupled. Improving the behavior of the MLS upper tropospheric composition observations in the presence of clouds was a primary goal of version 4. To accomplish that, the way in which individual limb spectra (of which there are 120 per vertical scan of the MLS antenna) were flagged as possibly being affected by thick clouds was redesigned, and the effects of these flags were changed to reject limb measurements both below and at the flagged location. These changes improved the handling of thick clouds, which have strong nonlinear effects on the observed radiances and must therefore be identified and rejected.

In addition, the representation of cloud structures in the atmosphere in the forward model and retrieval algorithm was changed, effectively increasing the vertical resolution. These changes improved the handling of thin cloud effects on the measured composition profiles, which can be represented directly and do not require measurements to be rejected.

These changes in how the retrieval algorithm handles clouds necessarily resulted in changes in the behavior of quantities used for data screening (quality, convergence, etc.); therefore, the post-processing flag thresholds were also changed. These changes are summarized in the v4 data documentation (Livesey et al., 2015), and are applied in our analysis of v4 data. Updates to the retrieval algorithm and the resulting changes in quality screening thresholds both contribute to increased data yields when using v4 relative to v3 in this region.

We have reorganized the text and added a few lines at the end of Section 2.2 to address this comment:

"The mean profiles shown in Fig. 2 are based on slightly different samples due to differences in the retrieval algorithm and quality control criteria. Specifically, v4 provides increased data yields in this region relative to v3 (10% more temperature profiles, 32% more water vapour profiles and 29% more ozone profiles). These increased data yields primarily reflect changes in the MLS quality screening criteria, which have been updated to account for changes in the way that clouds are handled during the retrieval step. One of the primary goals of MLS v4 was to improve the behavior of MLS upper tropospheric composition retrievals in the presence of clouds. This was accomplished by redefining the manner in which clouds were represented in the MLS forward model, and by redesigning the method by which the strongest cloud signals in the MLS radiances are flagged and excluded from the retrievals. These changes significantly reduce the sensitivity of the MLS composition observations to cloud scattering signals. Relative differences between v3 and v4 are effectively unchanged when the comparison is limited to retrievals that meet quality control criteria in both v3 and v4."

**(Fig 2) It is very difficult to see the uncertainties in these plots as presented, since the magnitudes of the mean values are much larger and hence mask the magnitudes of the uncertainties. Consider plotting the uncertainties separately or instead of the means, if that is the focus of the plot. In addition, why average together different years and different seasons into one plot?**

This plot is meant to serve as context for the plots of mean / median / RMSE bias plots to follow, and we intended the mean profiles and not the uncertainties to be the focus of the plot. The locations where differences exceed the combined uncertainties are described in detail in the text; however, for completeness, we will rearrange the plot to include the v4–v3 differences (including combined uncertainties) on separate narrow axes attached to each panel.

The different years, seasons and regions that are used in the average correspond to the four measurement campaigns, and mirror the combination of the four campaigns in the following plots of average biases. This approach is meant to capture a sample like "profiles that were or could have been included in the validation".

**(Fig 4) It is difficult to see if any differences exist between the error bars on the**

**mean (set as 2-SEM) and the error bars on the median (set as the IQR) in both (a) and (b). Consider offsetting them in the vertical.**

This issue seems to be due to an incomplete description of the figure in the caption. The error bars on the mean are represented by a shaded blue envelope, which is continuous from the bottom to the top of the profile. Values of the standard errors at MLS levels can be inferred by following variations in the thinner blue lines (at the edges of the envelope) between levels. This vertical continuity in the mean+standard error profiles is meant to serve the same purpose as a vertical offset. The error bars on the median are shown only at the MLS levels, and are consistently larger than the error bars on the mean (in this figure, though this is not necessarily true in other figures). We have updated the caption for this figure to clarify that uncertainties in the mean biases are shown as blue shaded envelopes bounded by thin blue lines.

**(Figs 7, 9, 10, 12) The "x" axes in these figures should be labelled with the tracer names, similar to how it was done in Figs 5 and 6, for quick visual reference.**

We agree, and have changed the figures accordingly — thanks for pointing this out.

**References**
Livesey, N. J., Read, W. G., Wagner, P. A., Froidevaux, L., Lambert, A., Manney, G. L., Millán-Valle, L. F., Pumphrey, H. C., Santee, M. L., Schwartz, M. J., Wang, S., Fuller, R. A., Jarnot, R. F., Knosp, B. W., and Martinez, E.: Version 4.2x Level 2 data quality and description document, Tech. Rep. JPL D-33509, NASA Jet Propulsion Laboratory, version 4.2x-1.0, 2015.

---

## Referee Comment (RC2) · Anonymous Referee #2 · 27 Apr 2016

The study by Yan et al. deals with the validation of Aura MLS temperature, water vapor and ozone retrievals over the Tibetan plateau during the Asian Summer Monsoon with light balloons in-situ observations. The validation methodology is thorough with the choice of MLS data according to the MLS team recommendations, the application of interpolation and the smoothing to the in-situ profiles. The focus of the paper is appropriate to AMT and the original balloon observations bring some interesting information about the MLS data. The comparison between MLS v3 and v4 data ar of particular

interest for the users. Nevertheless, I have two major comments to be dealt with before this study could be published in AMT.

The first comment concerns the added-value of the paper. The authors cite a number of previous publications (p2 l27-32) where MLS data have been validated globally and justify their study by the fact that none has focused on the Tibetan plateau region where the presence of the Asian Monsoon Anticyclone in the UTLS makes it particular relative to other UTLS regions. Nevertheless, the results are not discussed enough in light of the previous validation studies cited in the introduction and one does not really see clearly wether MLS retrievals have particular difficulties in reproducing Temperature, water vapor and ozone over this region during the monsoon, that is the added value of this study. The discussion part of the paper should therefore more clearly show how the results over the Tibetan plateau agree or disagree with the previous validation studies.

The second comment is concerning the presentation of the results. The profile figures are good and informative. The detailed statistics are rather difficult to follow and heavy to read because presented in a very descriptive way. Three different parameters and four different sites makes a large amount of numbers which are repeatedly presented all along the paper. Furthermore, a lot of information is present in the profile figures and does not need to be described in details in the text. For the pressure-weighted mean (differnet from the profils) the statistics should be presented in a more synthetic way. Furthermore, the paper mostly dicusses biases. The variability from MLS and the radiosonde are compared and the correlation between both are discussed in the text but not thoroughfully enough. They should appear in a more concise and synthetic way with Taylor diagramms complemented by the correponding numbers presented in tables (biases, biases of the RMs, RMS of sondes and MLS, correlation coefficients). Taylor diagramms are indeed the best way to synthetically compare the variabilities of different datasets and their correlations. With such diagramms and tables, the reader could see the agreement between both datasets in terms of correlation, RMS of the

biases and variabilities much more easily.

---

## Author Comment (AC2) · 21 Jun 2016

Our thanks to Anonymous Referee 2 for thoughtful comments and suggestions that have helped to improve the presentation in this manuscript. Our responses and a brief summary of related changes to the manuscript are given below.

**The first comment concerns the added-value of the paper. The authors cite a number of previous publications (p2 l27-32) where MLS data have been validated globally and justify their study by the fact that none has focused on the Tibetan plateau region where the presence of the Asian Monsoon Anticyclone in**

[Figure]

**the UTLS makes it particular relative to other UTLS regions. Nevertheless, the results are not discussed enough in light of the previous validation studies cited in the introduction and one does not really see clearly wether MLS retrievals have particular difficulties in reproducing Temperature, water vapor and ozone over this region during the monsoon, that is the added value of this study. The discussion part of the paper should therefore more clearly show how the results over the Tibetan plateau agree or disagree with the previous validation studies.**

We have reorganized and clarified the text in a few locations to better emphasize the added value relative to previous studies. The most important difference between our results and those reported previously is the sharp peak in the relative bias of ozone retrievals in the lowermost stratosphere, which does not appear in previous validations of MLS ozone. Other differences include larger dry biases and larger cold biases in the upper troposphere than have been reported by previous studies. Both the sharp peak in the ozone relative bias and the cold bias in the upper troposphere are enhanced during the monsoon relative to before monsoon onset, indicating that conditions in the Asian monsoon anticyclone pose unique challenges for remote sensing that affect the accuracy of MLS retrievals in this region. We find no significant difference in the upper tropospheric dry bias before and after monsoon onset. These additions supplement a series of notes in the discussion and summary sections of the original manuscript on how our results differ from those reported by previous studies.

**The second comment is concerning the presentation of the results. The profile figures are good and informative. The detailed statistics are rather difficult to follow and heavy to read because presented in a very descriptive way. Three different parameters and four different sites makes a large amount of numbers which are repeatedly presented all along the paper. Furthermore, a lot of infor-mation is present in the profile figures and does not need to be described in details in the text. For the pressure-weighted mean (differnet from the profils) the statistics should be presented in a more synthetic way. Furthermore, the**

**paper mostly dicusses biases. The variability from MLS and the radiosonde are compared and the correlation between both are discussed in the text but not thoroughfully enough. They should appear in a more concise and synthetic way with Taylor diagramms complemented by the correponding numbers presented in tables (biases, biases of the RMs, RMS of sondes and MLS, correlation coefficients). Taylor diagramms are indeed the best way to synthetically compare the variabilities of different datasets and their correlations. With such diagramms and tables, the reader could see the agreement between both datasets in terms of correlation, RMS of the biases and variabilities much more easily.**

We appreciate this suggestion, and have added three new figures and one table to address it. The three figures are modified Taylor diagrams showing standard deviations (normalized to sonde standard deviations), correlations, and RMS errors for the layer average values in the upper troposphere, tropopause layer, and stratosphere. Results are shown for both v3 and v4. Table 2 lists mean bias, RMS bias, bias of RMSs, and correlation for the same layers for all three variables. The text has been reorganized to reflect these additions, with the presentation modified to be more qualitative and less quantitative (since quantitative information is now provided in Table 2).